# An Electrophysiological and Proteomic Analysis of the Effects of the Superoxide Dismutase Mimetic, MnTMPyP, on Synaptic Signalling Post-Ischemia in Isolated Rat Hippocampal Slices

**DOI:** 10.3390/antiox12040792

**Published:** 2023-03-24

**Authors:** Martina Puzio, Niamh Moreton, Mairéad Sullivan, Caitriona Scaife, Jeffrey C. Glennon, John J. O’Connor

**Affiliations:** 1UCD School of Biomolecular & Biomedical Science, University College Dublin, Dublin 4, Ireland; 2Mass Spectrometry Core Facility, UCD Conway Institute of Biomolecular & Biomedical Research, University College Dublin, Dublin 4, Ireland; 3UCD School of Medicine, University College Dublin, Dublin 4, Ireland

**Keywords:** hypoxia, oxygen–glucose deprivation, hippocampus, EPSP, superoxide dismutase, ischemia-reperfusion injury, oxidative stress, synaptic transmission, synaptic plasticity, proteomics, MnTMPyP

## Abstract

Metabolic stress and the increased production of reactive oxygen species (ROS) are two main contributors to neuronal damage and synaptic plasticity in acute ischemic stroke. The superoxide scavenger MnTMPyP has been previously reported to have a neuroprotective effect in organotypic hippocampal slices and to modulate synaptic transmission after in vitro hypoxia and oxygen–glucose deprivation (OGD). However, the mechanisms involved in the effect of this scavenger remain elusive. In this study, two concentrations of MnTMPyP were evaluated on synaptic transmission during ischemia and post-ischemic synaptic potentiation. The complex molecular changes supporting cellular adaptation to metabolic stress, and how these are modulated by MnTMPyP, were also investigated. Electrophysiological data showed that MnTMPyP causes a decrease in baseline synaptic transmission and impairment of synaptic potentiation. Proteomic analysis performed on MnTMPyP and hypoxia-treated tissue indicated an impairment in vesicular trafficking mechanisms, including reduced expression of Hsp90 and actin signalling. Alterations of vesicular trafficking may lead to reduced probability of neurotransmitter release and AMPA receptor activity, resulting in the observed modulatory effect of MnTMPyP. In OGD, protein enrichment analysis highlighted impairments in cell proliferation and differentiation, such as TGFβ1 and CDKN1B signalling, in addition to downregulation of mitochondrial dysfunction and an increased expression of CAMKII. Taken together, our results may indicate modulation of neuronal sensitivity to the ischemic insult, and a complex role for MnTMPyP in synaptic transmission and plasticity, potentially providing molecular insights into the mechanisms mediating the effects of MnTMPyP during ischemia.

## 1. Introduction

Stroke is a leading cause of morbidity and mortality worldwide, with ischemic stroke being the most common type of stroke [1]. One of the most devastating consequences of ischemic stroke is the impairment of synaptic plasticity which contributes to the cognitive deficits that occur after an insult [1,2]. While the recanalization of the blood flow remains the main therapeutic approach, reperfusion after ischemia can lead to further damage, termed ischemia-reperfusion injury (I-RI) [3]. One of the key mechanisms by which IRI causes injury is through the effects of oxidative stress [3,4]. During ischemia, the release of the excitatory neurotransmitter glutamate overstimulates the ionotropic glutamate receptors, causing a sustained increase in intracellular calcium. The main pathway is triggered by N-methyl-D-aspartate (NMDA) receptors, leading to a number of downstream deleterious effects, including alteration in mitochondrial electron transport and an increase in reactive oxygen species (ROS) production [5,6].

ROS, such as superoxide (O_2_), hydrogen peroxide (H_2_O_2_) and hydroxyl, are highly reactive molecules that can directly damage proteins, lipids, nucleic acid and other cellular components [7]. Subcellular organelles display complex proteomic responses to impaired energy metabolism and oxidative stress, with these responses ultimately determining the death or survival of the cell. Proteomic analysis by other researchers has demonstrated the activation of neuronal pathways following ischemia and reperfusion, including mitochondrial energy production, endoplasmic reticulum stress and ribosomal dysfunction [8]. This metabolic stress also induces microglial release of remodelling factors and cytokines, such as VEGF and TGF-β1. A plethora of cellular activities, including cell proliferation, differentiation, apoptosis, adhesion and migration, are controlled by TGF-β. It also plays an important role in the regulation of glucose metabolism, modulating glucose transport type 1 (Glut1) mRNA levels and increasing glucose uptake [9]. Bae et al. outlined a role for this cytokine in hippocampal synaptic transmission and plasticity. They reported that TGF-β1 regulates the expression and localization of key subunits of α-amino-3-hydroxy-5-methyl-4-isoxazoleproprionic acid (AMPA) and NMDA receptors, influencing glutamate-evoked currents and facilitation of neurite outgrowth [10]. The TGF-β signalling pathways are regulated by the plasticity and dynamicity of the actin cytoskeleton, which alters the distribution and activity of ligands and receptors [11].

Recent reports have implicated changes to the dynamics of the actin cytoskeleton upon the release of ROS from mitochondria and subsequent cell death. The actin cytoskeleton is involved in the generation and maintenance of cell morphology, in endocytosis, intracellular trafficking and in cell division [12]. In neurons, about 80% of actin is in a dynamic state, and its turnover can account for 50% of ATP utilization [13]. Therefore, the actin cytoskeleton is impacted by ATP depletion conditions, such as ischemia. The alteration of the structure of the actin cytoskeleton causes a redistribution of NMDA receptors from dendritic spines and reduces the activity of the receptors at the synapse [14]. A pro-oxidant state is also involved in vesicular trafficking, modulating the synaptic distribution of NMDA and AMPA receptors, crucial mediators of synaptic transmission and potentiation [15,16].

Modulation of the cellular oxidant state using scavengers has been reported to affect ROS-mediated receptor trafficking and dysfunctions. In a model of ischemia/reperfusion injury, the superoxide dismutase mimetic MnTMPyP prevented the internalization and subsequent endocytic trafficking of synaptic receptors [17]. In an in vivo study in mice exposed to brief and repetitive episodes of I-RI, the administration of MnTMPyP rescued synaptic plasticity, ameliorating the deleterious effect of the hypoxic insult on synaptogenesis [18,19]. We have previously shown that the superoxide dismutase mimetic MnTMPyP has a neuroprotective effect in hippocampal slices exposed to hypoxia and oxygen–glucose deprivation (OGD) [20].

To further explore its biological effects on synaptic transmission and plasticity, we tested whether MnTMPyP affects LTP post-hypoxia and OGD in the CA1 area of the hippocampus. Additionally, we performed mass-spectrometry-based proteomic analysis in the hippocampus (followed by protein enrichment analyses) to assess the cellular adaptive responses to metabolic stress and how these are modulated by MnTMPyP.

## 2. Materials and Methods

All animal experiments were approved by the Animal Research Ethics Committee of the Biomedical Facility at University College Dublin (protocol code AREC-20-30-Oconnor, 5 January 2021) in accordance with the European legislation.

### 2.1. Preparation of Acute Hippocampal Slices

Three-to-four-week-old male and female Wistar rats (p21–p28) were used in these experiments. The rats were sourced from Charles River and bred in the Biomedical Facility in UCD. Animal diet was a Global Diet 2918. Animals were anesthetized with 5% isoflurane and euthanized by decapitation using a guillotine. The brain was rapidly removed and mounted for vibratome sectioning in ice-cold dissection artificial cerebrospinal fluid (aCSF) consisting of 125 mM NaCl, 25 mM NaHCO_3_, 1.25 mM NaH_2_PO_4_, 2.5 mM KCl, 10 mM glucose, 5 mM MgSO_4_ and 1 mM CaCl_2_ and perfused with 95% O_2_/5% CO_2_. The 400 µm transverse hippocampal slices were cut using a Leica VT 1000 S Vibroslice. Slices were immediately transferred into a holding chamber containing recording aCSF, consisting of 125 mM NaCl, 25 mM NaHCO_3_, 1.25 mM NaH_2_PO_4_, 2.5 mM KCl, 10 mM glucose, 1.3 mM MgSO_4_ and 2.5 mM CaCl_2_ perfused with 95% O_2_ and 5% CO_2_ for 1 h at room temperature. Hippocampal slices were then transferred to a submerged recording bath continuously perfused with recording aCSF from a 50 mL reservoir (30–31 °C) at a flow rate of 4 mL/min and bubbled with 95% O_2_/5% CO_2_. Slices were allowed to adjust to the environment of the recording bath for at least 30 min before recording.

### 2.2. Electrophysiology

Field excitatory postsynaptic potentials (fEPSPs) were elicited by stimulation of the Schaffer collateral pathway of the CA1 region in the hippocampus using aCSF-filled monopolar glass electrodes (1B150F-4, World Precision Instruments). fEPSPs were recorded from the CA1 pyramidal neurons. The stimulating electrode was connected to a S48 Stimulator (Grass Instruments; A-M Systems, Sequim, WA, USA) via a Grass SIU5 stimulus isolation unit. The recording electrode was connected to a P55 AC-coupled amplifier (Grass Instruments; AstroNova, West Warwick, RI, USA) and fEPSPs were acquired at 20 kHz. Stimulus strength was adjusted in order to give 40% of the maximal response, determined by input/output curves. Paired fEPSPs were stimulated every 30 s, separated by 50 ms intervals. Hippocampal slices were stimulated for a minimum of 20 min at 40% of maximal fEPSP slope in order to obtain a stable baseline before drug application or exposure to hypoxic condition. In order to evaluate the effects of the drug on baseline synaptic transmission, the compound was applied after 20 min baseline and incubated for 1 h. In long-term potentiation (LTP) experiments, synaptic potentiation was induced by high-frequency stimulation (HFS) consisting of 3 trains of 1 s duration every 30 s at 100 Hz. In post-ischemic LTP experiments, control and treated hippocampal slices were exposed to 20 min hypoxia or 10 min OGD. After 40 min of recovery, LTP was evoked using the same HFS protocol as previously described. Slices were incubated with MnTMPyP (25 µM and 2.5 µM) for 20 min before switching to ischemic conditions and LTP induction. Recordings were acquired and analysed using the software package WCP (J. Dempster, Strathclyde).

### 2.3. Ischemia-Reperfusion Injury

Hypoxic slices were exposed to hypoxia by switching the gas that passed over the slice chamber from 95% O_2_/5% CO_2_ to 95% N_2_/5% CO_2_ for 20 min. Re-oxygenation was obtained by switching the gas from 95% N_2_/5% CO_2_ back to 95% O_2_/5% CO_2_. Previous reports in our laboratory have reported O_2_ brain slice levels that occur from this procedure [21,22]. OGD slices were perfused with glucose-free aCSF supplemented with equimolar sucrose and gassed with nitrogen for 10 min (95% N_2_/5% CO_2_). At the end of the OGD episode, slices were reperfused with oxygenated-glucose-containing aCSF. Slices were incubated with MnTMPyP (25 µM and 2.5 µM) for 20 min before performing hypoxia or OGD.

### 2.4. Liquid Chromatography Tandem Mass Spectrometry (LC-MS/MS)

#### 2.4.1. Sample Treatment for LC-MS/MS

Three-to-four-week-old male and female Wistar rats (p21–p28) were used in these experiments. Animals were anesthetized and euthanised, and hippocampal slices (400 µM) were extracted as previously described (Section 2.1). Slices were treated for LC-MS/MS on the electrophysiology rig where one *n* represents four hippocampal slices from the same animal. Treatment groups included: 1. normoxia; 2. hypoxia, 3. OGD, 4. MnTMPyP alone; 5. MnTMPyP and hypoxia; and 6. MnTMPyP and OGD, where each treatment group contained *n* = 4 (each *n* is from a different animal). For controls (normoxia), slices were perfused for 2 h with 95% O_2_/5% CO_2_. For the hypoxia group, slices were perfused with (95% N_2_/5% CO_2_) for 20 min followed by 40 min reperfusion (95% O_2_/5% CO_2_). For the OGD group, slices were perfused with 95% N_2_/5% CO_2_ and glucose-free aCSF followed by 40 min reperfusion (95% O_2_/5% CO_2_). For MnTMPyP-treated groups, slices were perfused with 95% O_2_/5% CO_2_ and MnTMPyP only for 2 h. For MnTMPyP and hypoxia/OGD-treated groups, slices were perfused with MnTMPyP for 1 h followed by 20 min hypoxia (95% N_2_/5% CO_2_) or 20 min OGD (95% N_2_/5% CO_2_ and glucose-free aCSF), followed by reoxygenation (95% O_2_/5% CO_2_) for 40 min. Samples were stored at −80 °C (in aCSF) before being prepared for LC-MS/MS.

#### 2.4.2. Digestion and Mass Spectrometry of Proteomic Samples

Each *n* sample of 4 hippocampal tissue slices (as described above) was first lysed in 200 μL Urea buffer (8 M Urea, 50 mM triethylammonium bicarbonate) containing protease inhibitors (Roche, cOmplete Mini, Cat Nr 11836153001) sonicated in 3 × 10 s bursts with a handheld probe (Microson Ultra Sonic Cell Disruptor XL, Misonix Inc. New York, NY, USA) while ensuring samples remained on ice between bursts, and then spun (15,000× *g* for 10 min at 4 °C) to remove cell debris. Supernatants were collected and protein content was determined using a modified Bradford assay (Ramagli and Rodriguez, Electrophoresis 1985, *6*, 559–563). For-digestion samples were diluted to 50 μg in 50 μL with 50 mM TEAB, reduced with dithiothreitol for 1 h (27 °C, final concentration 10 mM) and alkylated for 30 min. with iodoacetamide (RT, final concentration 20 mM). Prior to addition of trypsin (1 μg, Pierce Trypsin Protease MS grade, Cat Nr 90057), samples were diluted with 50 mM TEAB (pH 8.0) to ensure the correct pH and to reduce the Urea content to <2 M. Digestion was carried out at 37 °C with agitation on a Thermomixer (16 h) and subsequently halted with the addition of TFA (trifluoroacetic acid) to a final concentration of 1%.

The tryptic peptides were desalted using ZipTips (C18, Millipore ZTC18S096) as per the manufacturer’s instructions and resuspended in 0.1% formic acid for mass spectrometry. Peptide content was measured on a DeNovix spectrometer to ensure equal loadings of samples for mass spectrometry. Desalted peptides were loaded onto EvoTips and run on a timsTOF Pro mass spectrometer (Bruker Daltonics, Bremen, Germany) coupled to the EvoSep One system (EvoSep BioSystems, Odense, Denmark). The peptides were separated on a reversed-phase C_18_ Endurance column (15 cm × 150 μm ID, C_18_, 1.9 μm) using the preset 30 SPD method. Mobile phases were 0.1% (*v*/*v*) formic acid in water (phase A) and 0.1% (*v*/*v*) formic acid in acetonitrile (phase B). The peptides were separated by an increasing gradient of mobile phase B for 44 min using a flow rate of 0.5 μL/min.

The timsTOF Pro mass spectrometer was operated in positive ion polarity with TIMS (Trapped Ion Mobility Spectrometry) and PASEF (Parallel Accumulation Serial Fragmentation) modes enabled. The accumulation and ramp times for the TIMS were both set to 100 ms with an ion mobility (1/k0) range from 0.6 to 1.6 Vs/cm. Spectra were recorded in the mass range from 100 to 1700 *m*/*z*. The precursor (MS) Intensity Threshold was set to 1000 and the precursor Target Intensity set to 20,000. Each PASEF cycle consisted of 1 MS ramp for precursor detection followed by 5 PASEF MS/MS ramps, with a total cycle time of 1.17 s.

#### 2.4.3. Data Analysis for Mass Spectrometry

Raw data from the timsTOF mass spectrometer were analysed with MaxQuant (v 2.0.3.0). The Uniprot Rat reference proteome (downloaded 17 June 2021) was searched against using the following parameters: Trypsin was selected as the digesting enzyme. Variable modifications were set at oxidation on methionine and N-terminal acetylation, while carbamidomethylation of cysteine was set as a fixed modification. Two missed cleavages were allowed. The LFQ (Label-Free Quantitation) option was selected, as was the ‘Match between runs’ option. A FASTA file containing common contaminants was also selected to be searched. Subsequent statistical analysis was carried out with Perseus software (v 2.0.6.0) using the LFQ Intensity values in the proteinGroups.txt MaxQuant output file. Data were first filtered to remove contaminants and reverse hits. The data were then log2 transformed and filtered such that proteins listed appeared in a minimum of 70% of all samples. Absent values were then replaced by imputation of values from the normal distribution. Samples were grouped according to treatment category. Two-way ANOVA tests, followed by post hoc two-sample unpaired two-tailed Student’s *t*-tests were then performed to determine proteins that were statistically differentially expressed between the treatment groups (*p* < 0.05). The mass spectrometry proteomics data were deposited in the ProteomeXchange Consortium via the PRIDE [1] partner repository with the dataset identifier PXD040890.

#### 2.4.4. Functional Enrichment Analysis

Ingenuity Pathway Analysis software (Qiagen, Hilden, Germany) was used to identify significantly enriched canonical pathways and networks in the above treatments, based on significantly differentially expressed proteins based on two-sample two-tailed unpaired Student’s *t*-tests. The significance of association between the given dataset and predicted canonical pathway was calculated using the right-tailed Fischer’s exact test with the Benjamini–Hochberg correction applied for multiple comparisons. For prediction of associated canonical pathways and enriched networks, Ingenuity Pathway Analysis software used the Ingenuity Knowledge Base, which is a repository of data based on extensive information from published literature. For canonical pathways, apart from the *p*-value of overlap, a ratio indicating the strength of the association was also provided (the number of genes from the dataset that map to the pathway divided by the total number of genes that map to the canonical pathway). Pathways with high ratios and low *p*-values may be the most likely candidates for an explanation of the observed phenotype. For each network, the Ingenuity software generates an enrichment score that takes into account the number of eligible molecules/proteins in the network and its size, as well as the total number of network-eligible molecules analysed and the total number of molecules in the Ingenuity Knowledge Base that could potentially be included in networks.

### 2.5. Drugs

Cell-permeant SOD mimetic manganese (III) tetrakis (1-methyl-4-pyridyl) porphyrin (MnTMPyP) was purchased from Enzo Life Sciences and dissolved in dimethyl sulfoxide (DMSO) obtained from Sigma. MnTMPyP was dissolved in DMSO with a final concentration equal or less than 0.1% when diluted in aCSF. DMSO control experiments were carried out at the same dilution. For LC-MS/MS, treatments with MnTMPyP were at a final concentration of 25 µM. For electrophysiology, treatments with MnTMPyP were at a final concentration of both 2.5 µM and 25 µM.

### 2.6. Statistical Analysis of Electrophysiological Studies

Electrophysiological recordings were acquired and analysed using the software package WinWCP (Ver 5.2.7, J. Dempster, Strathclyde, UK). All fEPSP slope measurements are represented as a percentage of their initial mean baseline slope. Baseline recordings were determined by the average of fEPSP slope over a 20 min period prior to drug application or hypoxia/OGD induction. All data are presented as mean ± SEM. Statistical analysis between controls and drug-treated slices was tested with an unpaired Student’s *t*-test using Prism Software (Ver 9.0 GraphPad). A minimum of *p* < 0.05 was considered to be statistically significant. The *n* values correspond to the number of slices used in each experimental group.

## 3. Results

### 3.1. The Effects of MnTMPyP on Synaptic Transmission and Plasticity

In the CA1 region, there was no change in fEPSP slope over a 1 h time-course (20 min: 103 ± 2.5%; 1 h: 98 ± 3.3%; *p* > 0.05; *n* = 19). To determine the preconditioning effect of the SOD mimetic, MnTMPyP (2.5 µM and 25 µM), slices were perfused with MnTMPyP for 1 h once a stable baseline had been established for 20 min. Pre-treatment with 2.5 µM MnTMPyP did not significantly alter baseline transmission (20 min: 105.3 ± 3.35%; 1 h: 81.9 ± 14.1%; *p* > 0.05; *n* = 11). However, in the presence of 25 µM MnTMPyP there was a significant decrease in fEPSP slope over 1 h (baseline 20 min: 105.9 ± 5.2%; baseline 1 h: 76.3 ± 7.5%; *n* = 11; *p* < 0.05) (Figure 1A,B). To determine the preconditioning effects of MnTMPyP (2.5 µM and 25 µM) on synaptic plasticity, slices were stimulated with HFS. Control slices gave rise to a robust synaptic potentiation (135.0 ± 6.1%, *n* = 27) after 60 min. Application of MnTMPyP (2.5 µM) did not impair synaptic plasticity (121.0 ± 9.2, *n* = 9). However, higher concentrations of MnTMPyP (25 µM) impaired synaptic potentiation and significantly reduced baseline synaptic transmission 60 min after the HFS (74.3 ± 9.3%, *n* = 11) (Figure 1C,D).

### 3.2. MnTMPyP Does Not Affect Synaptic Transmission during OGD and Post-Ischemic LTP

It has previously been reported that application of MnTMPyP (25 µM) modulates synaptic transmission recovery after an insult of 20 min OGD in the CA1 area of the hippocampus [20]. In order to evaluate the effect of the superoxide scavenger on post-ischemic synaptic plasticity, a 10 min model of oxygen–glucose deprivation was used. This model allows a full recovery of fEPSP after the ischemic episode and synaptic potentiation following HFS. In control slices, a 10 min OGD episode gave rise to a maximum depression of fEPSP to 48.0 ± 10.0% and a recovery to 99.5 ± 10.1%, (*n*  =  5) (Figure 2A,B). Application of MnTMPyP (25 µM) did not affect the fEPSP depression (16.4 ± 4.2%) but had an inhibitory effect during the recovery (35.0 ± 11.4%), similar to what was reported previously. To investigate whether the effect was reversible, a lower concentration of the superoxide scavenger was tested. MnTMPyP (2.5 µM) did not modulate the OGD-induced depression or recovery of the fEPSP (17.0 ± 3.4% and 113.0 ± 3.0%, respectively) (Figure 2A,B). In order to test the effect of the compound on synaptic potentiation, LTP was induced in slices previously exposed to 10 min OGD. Ischemic slices gave rise to a stable LTP over 60 min (131.0 ± 6.1%; *n* = 7). The magnitude of the post-ischemic synaptic strengthening was not different from LTP induced in normoxic slices (135.0 ± 6.1%, *n* = 27) (Figure 2C,D). To evaluate the effect of 2.5 µM MnTMPyP on LTP post-OGD, slices were pretreated for 20 min with the scavenger and exposed to OGD. At the end of the 40 min recovery, HFS was induced to evoke LTP. The stimulation gave rise to a synaptic potentiation which was not different in magnitude compared to control LTP post-OGD (129.5 ± 7.5%, *n* = 5) (Figure 2E,F).

### 3.3. Application of MnTMPyP Modulates Synaptic Transmission and Plasticity Post-Hypoxia

In the CA1 region, 20 min hypoxia gave rise to a maximum depression of fEPSP slope to 32.45 ± 6.8% and full recovery (135.41 ± 12%) upon reoxygenation (*n* = 9). To determine the preconditioning effect of MnTMPyP (2.5 µM), slices were perfused with MnTMPyP 20 min prior to the hypoxic episode. Pre-treatment with MnTMPyP significantly decreased the fEPSP slope recovery 40 min post-hypoxia (66.24 ± 8%; *n* = 8, *p* < 0.05; Figure 3A,B). To determine whether hypoxia had a preconditioning effect on LTP, slices were exposed to 20 min hypoxia and allowed to recover for 40 min prior to HFS. HFS induced a stable LTP in control slices lasting 60 min (135.0 ± 6.1%, *n* = 27) with no difference compared to LTP induced in slices pre-exposed to hypoxia (144.87 ± 5.67%; *n* = 4), suggesting a 20 min hypoxic episode does not modulate LTP (Figure 3C,D). To examine whether 2.5 µM MnTMPyP application had an effect on LTP post-hypoxia, slices were pre-treated with 20 min hypoxia and HFS was induced 40 min after the hypoxic episode. Here, the stimulation gave rise to a stable synaptic potentiation lasting for 60 min (144.87 ± 5.67%; *n* = 4). However, application of 2.5 µM MnTMPyP significantly impaired LTP post-hypoxia (113.4 ± 11.9%; *n* = 4, *p* < 0.05) (Figure 3E,F).

### 3.4. Post-Hypoxia Recovery Yields Upregulation of Synaptogenesis, Phagocytosis and Cell Stress Response

Proteomic analysis of CA1 regions following a hypoxic episode showed that the greatest fold change compared to control normoxia-treated slices concerned the downregulation of *Adgrl1*, responsible for functional synapse formation and maturation [23], in addition to the downregulation of *Kctd4*, responsible for potassium channel tetramerization, with the literature recently implicating the Kctd family of proteins in autism and schizophrenia via its role in the regulation of actin cytoskeleton dynamics [24]. The most significant change concerns the upregulation of *Hgs*, a regulator of endosomal trafficking and lysosomal-mediated protein degradation, followed by the upregulation of *Uqcrh*, a subunit of the mitochondrial electron transport chain (Figure 4A). Network enrichment analysis of significantly differentially expressed targets generated the most enriched network (*p* = 1.00 × 10^−52^) with dramatic upregulation of actin at its centre, suggesting marked upregulation of synaptogenesis and dendrite growth (Figure 4B). Actin is a major cytoskeletal component responsible for both the establishment and maintenance of neuronal structure and plays essential roles in appropriate dendritogenesis [25]. Other central nodes of this network include upregulation of *ACTB* (Beta-actin), upregulation of the BCR complex (antigen presentation signalling) and upregulation of *Hsp90* (cell stress). Phagocytosis is hypothesised to increase following hypoxia as a compensatory mechanism against elevated vulnerability to bacterial invasion [26]. Ingenuity canonical pathway analysis based on highlighted targets confirms this network, with the most significant signalling pathways implicating actin (*p* = 5.3 × 10^−5^) cytoskeletal dynamics (including Integrin (*p* = 2.2 × 10^−5^), which interacts with actin for cytoskeletal stabilisation) and phagocytosis (*p* = 5.2 × 10^−7^) (Table 1).

### 3.5. Post-OGD Recovery Promotes Synaptogenesis, Microtubule Function and Mitochondrial Function

A volcano plot (of protein expression changes plotted against their respective *p* values) generated following OGD recovery in CA1 regions shows *Ensa* as the target with greatest fold change (downregulated), which regulates ATP-governed K+ channels (Figure 5A). Neurogranin (*Nrgn)* is also downregulated, and is typically a positive regulator of LTP [27]. The most significantly changed target is the downregulation of *Ctsb*, responsible for appropriate lysosomal-mediated proteolysis and autophagy. *Akap5* (downregulated) anchors protein kinase A (PKA) to the postsynaptic membrane and thus regulates AMPA-mediated synaptic plasticity [28]. Amongst the most significant upregulated targets are ribosomal proteins (*Rpl32*, *Rps9*), and the regulator of endosome trafficking *Rab5c*, with reported roles in promoting dendritic branching [29]. Ingenuity network enrichment analysis shows the top enriched network (*p* = 1.00 × 10^−48^) with actin as a central node; however, directionality is undetermined (Figure 5B). Other central nodes include the dramatic upregulation of *CAMKII*, with roles in modulating glutamatergic synaptic plasticity and LTP [30], in addition to mitochondrial maintenance [31]. Beta-tubulin, a component of microtubules, necessary for the transport of organelles and vesicles throughout the neuron, is also heavily upregulated. This indicates increased synaptic plasticity signalling, particularly in glutamatergic neurons, in addition to elevated intra-neuronal vesicular transport in OGD recovery. The top predicted pathways according to ingenuity pathway analysis concern PKA signalling (*p* = 5.3 × 10^−5^), necessary for synaptic plasticity, in addition to ARE-mediated mRNA turnover (*p* = 5.8 × 10^−5^), with reported roles in the degradation of transcripts produced during environmental insult [32] and mitochondrial function (*p* = 9.5 × 10^−5^) (Table 2).

### 3.6. MnTMPyP in Baseline Normoxia Protects against Mitochondrial-Mediated Apoptosis, but Impairs Synaptogenesis

Volcano plot visualisation of important targets shows the dramatic downregulation of three proteins: *Scamp3* is responsible for endocytosis and membrane trafficking, typically concentrated in synaptic vesicles. *Nit1* is responsible for the hydrolysis of the antioxidant glutathione. Downregulation of the postsynaptic protein kinase Neurogranin (*Nrgn)* may mediate impaired synaptic development and remodelling [27] (Figure 6A). Network analysis of significantly differentially expressed proteins yielded two networks of equal enrichment score (*p* = 1.00 × 10^−48^) (Figure 6B). The first of these networks has *VDAC1* (downregulated) and *RAB7A* (downregulated) at its centre. *VDAC1* functions as a gatekeeper of mitochondria–cell metabolites, and interacts with mitochondria-mediated apoptosis components, namely Bcl-2, to promote apoptosis [33]. *RAB7A* mediates the transport of endosomes within the axon and within mitochondria. Another important node is the downregulation of *Tomm40*, a mitochondrial translocase that mediates protein import through the outer mitochondrial membrane and contributes to the assembly of the mitochondrial membrane respiratory chain. The second network has *MAPT* (tau) downregulated at its centre. Tau functions to regulate neuronal structure and polarity via promoting microtubule assembly. Interestingly, RAB7A may promote tau secretion, indicating connectivity between the top two networks [34]. Combining these results into ingenuity canonical pathway analysis (Table 3) shows synaptogenesis signalling to be the most significant (*p* = 6.3 × 10^−8^), followed by the cell-stress-associated chaperone family BAG2 (*p* = 1.6 × 10^−7^) and mitochondrial dysfunction (*p* = 1.2 × 10^−6^).

### 3.7. MnTMPyP May Decrease Synaptogenesis While Also Reducing Mitochondrial-Mediated Apoptosis and Cell Stress following Hypoxia

A volcano plot to visualise data shows *Kctd4* as the protein with the most upregulation in treatment groups, implicated in potassium channel tetramerization, converse to findings in hypoxia recovery without MnTMPyP above (Figure 7A). The literature has implicated the *Kctd* family of proteins in autism and schizophrenia via its role in the regulation of actin cytoskeleton dynamics [18]. LOC100910056, also known as *Bltp3b*, is dramatically downregulated, and enables GARP complex binding, necessary in the appropriate trafficking of vesicles from endosomes [35]. The most significantly upregulated protein, Histidyl tRNA Synthetase (Hars), plays a role in axonal guidance. Network analysis of significantly changed protein expression following MnTMPyP treatment of the hypoxic CA1 region revealed the most enriched network (*p* = 1.00 × 10^−52^) with actin at its centre, interacting with eight other proteins, indicating signalling for appropriate dendritogenesis (Figure 7B). Directionality was no longer highly upregulated, as in hypoxia recovery in the absence of the drug. Another central node, *ROCK*, is a regulator of actin organisation [36]. Other central nodes indicate a neuroprotective role of MnTMPyp, with downregulation of heat shock proteins *HSP90* (and its regulator *HDAC6*) and *HSP90AA1*, an opposing result to hypoxia recovery without the drug. This may suggest that ROS depletion by the drug abolishes the need for upregulation of heat shock proteins. Additionally, among the nine enriched canonical pathways (*p* < 0.001), two concern actin polymerisation and adhesion (Actin Cytoskeleton Signalling; *p* = 1.2 × 10^−4^, Integrin Signalling; *p* = 1.2 × 10^−3^). Other highlighted pathways concern metabolic and apoptosis regulation (Mitochondrion-expressed Sirtuin Signalling pathway; *p* = 1.2 × 10^−8^, Granzyme A signalling; *p* = 5.7 × 10^−4^) and cellular stress response (BAG2 Signalling pathway; *p* = 3.3 × 10^−5^) (Table 4).

### 3.8. MnTMPyP Reduces Synaptogenesis and Mitochondrial-Mediated Apoptosis following OGD

A volcano plot of differentially expressed targets following MnTMPyP treatment of CA1 regions after oxygen and glucose deprivation highlights the dramatic upregulation of Neuronal Growth regulator 1 (*Negr1*) in treated samples, necessary for regenerative axon sprouting (Figure 8A). Amongst the most significantly changed proteins is *Tenm4* (downregulated), responsible for the regulation of myelination of axons and of appropriate neuronal connectivity during development, and *Ndrg2* (upregulated) implicated in neurite outgrowth. Further characterisation using network enrichment analysis of significantly differentially expressed proteins shows the most enriched network (*p* = 1.00 × 10^−42^) with severely downregulated *TGFB1* at its centre, in addition to *AKT1* (Figure 8B). *TGFB1* has a multitude of roles, including cell growth, proliferation, differentiation and apoptosis, and is protective against neuronal degeneration and excitotoxic injury [37]. Interestingly, *TGFB1* negatively regulates mitochondrial respiration, with its downregulation in the top network suggesting increased mitochondrial bioenergetics [38]. Another central node is the upregulation of *CDKN1B*, a regulator of cell-cycle progression, indicating reduced cellular proliferation following MnTMPyP administration. Downregulation of the proto-oncogene *Myc* suggests reduced cell proliferation and reduced apoptosis due to its regulatory role in mitochondrial *Bcl-2* [39]. Three canonical pathways reach the significance threshold of *p* < 0.001 concerning mitochondrial-mediated apoptosis (Granzyme A signalling, *p* = 0.1 × 10^−3^) and respiration (oxidative phosphorylation, *p* = 8.7 × 10^−3^), in addition to Rho–actin interactions (*p* = 9.3 × 10^−3^), essential for actin polymerisation in neurogenesis, in addition to macrophage/microglia morphology (Table 5).

### 3.9. Two-Way ANOVA Analysis across All Groups in Hypoxia and OGD Experiments Confirms Alterations in Synaptogenesis, Cell Stress and Mitochondrial Function, with Heightened Sensitivity of Hypoxic CA1 Slices to MnTMPyP

A two-way ANOVA analysis comparing all groups in the hypoxia experiment reconfirmed the significance of actin signalling changes, with the post hoc Student’s *t*-tests demonstrate its upregulation following hypoxia. Actin is a major scaffold protein involved in neuronal structural integrity and growth. Cross-group ANOVA analysis similarly showed significant upregulation in hypoxia recovery (normoxia vs. hypoxia (F(1,12) = 5.43; normoxia vehicle vs. hypoxia vehicle, post hoc *t*-test t(12) = 2.71, *p* = 0.019 (Figure 9). The drug MnTMPyP upregulated actin-mediated signalling in normoxia (vehicle vs. MnTMPyp F(1,12) = 16.18; normoxia vehicle vs. normoxia MnTMPyp, post hoc *t*-test t(12) = 3.91, *p* = 0.0021) but not hypoxia, reflecting Figure 6B and Figure 7B previously. Cell stress changes have been implicated above in the hypoxia and MnTMPyP response, with the histone deacetylase HDAC6, an inducer of the already-highlighted heat shock protein HSP90 [40], showing no significant change between normoxia and hypoxia treatment. However, HDAC6 shows significant downregulation following MnTMPyP in hypoxia (vehicle vs. MnTMPyP F(1,12) = 0.15; hypoxia:vehicle vs. hypoxia:MnTMPyP post hoc *t*-test t(12) = 2.19; *p* = 0.048), and is a central node in the enriched network in Figure 7B. This indicates a potential mechanism through which the drug is protective via alleviating the cell stress response. Indeed, ROS has previously been shown to induce HDAC6 expression, with alleviation of oxidative stress following MnTMPyP administration being a possible means through which the cell stress response is abated [41]. Another target that was significantly upregulated following hypoxia is Uqcrh, forming a subunit of the mitochondrial electron transport chain. This shows significant upregulation in hypoxia recovery (normoxia vs. hypoxia F(1,12) = 5.08; normoxia:vehicle vs. hypoxia:vehicle post hoc *t*-test t(12) = 2.20; *p* = 0.048), in agreement with what was highlighted in Figure 4A, and is upregulated following MnTMPyP treatment in normoxia conditions (vehicle vs. MnTMPyP F(1,12) = 8.19; normoxia:vehicle vs. normoxia:MnTMPyP post hoc *t*-test t(12) = 2.63; *p* = 0.022). This may reflect a mechanism through which the drug, administered in unstressed conditions, may improve mitochondrial function. OGD treatment induced significant upregulation of CAMK2A in recovery (normoxia vs. OGD F(1,12) = 0.63; normoxia:vehicle vs. OGD:vehicle post hoc *t*-test t(12) = 4.04; *p* = 0.0016), in agreement with Figure 5A above. This is a major regulator implicated in modulating glutamatergic synaptic plasticity and LTP [30], in addition to mitochondrial maintenance [31], and is a central node in Figure 5B above. MnTMPyP generates no change to CAMK2A expression. RAB5C, a target implicated in endosome-mediated waste clearance and dendritic branching, shows significant upregulation in OGD recovery (normoxia vs. OGD F(1,12) = 15.46; normoxia:vehicle vs. OGD:vehicle post hoc *t*-test t(12) = 3.6; *p* = 0.0035), suggesting elevated synaptic growth. MnTMPyP has no obvious effect on this protein expression. Ctsb, a regulator of lysosomal-mediated proteolysis and autophagy, and inducer of apoptosis [42], shows downregulation in OGD recovery (normoxia vs. OGD F(1,12) = 31.49; normoxia:vehicle vs. OGD:vehicle post hoc *t*-test t(12) = 5.59; *p* = 0.0001), with MnTMPyP showing a potentially protective downregulating effect in normoxia conditions (vehicle vs. MnTMPyP F(1,2) = 2.99; normoxia:vehicle vs. normoxia:MnTMPyP post hoc *t*-test t(12) = 2.84; *p* = 0.015). Overall, MnTMPyP had a reduced influence on expression of these key proteomic targets following OGD, compared to hypoxia, reflecting the electrophysiological results above.

## 4. Discussion

Despite the vast research carried out on the neuroprotective role of antioxidants in counteracting oxidative stress, there is still a gap in the literature on how exactly they modulate synaptic transmission and plasticity [6]. The modulatory effects of antioxidant treatment with MnTMPyP on synaptic transmission and neuronal viability post-ischemia have been previously investigated [20]. In these studies, a high concentration of MnTMPyP (25 µM) induced a decreased fEPSP recovery post-ischemia in the CA1 region of the hippocampus. In addition, evidence suggests that these electrophysiological effects were neuroprotective, using cell viability analysis on organotypic slice cultures. In addition, the modulatory effects of MnTMPyP on synaptic plasticity were examined post-ischemia in the hippocampus (CA1) using two distinct in vitro ischemic models: OGD and hypoxia. Interestingly, distinct differences in the sensitivity of these models in their ability to achieve post-ischemic LTP with MnTMPyP treatment were observed. Prior treatment with a low concentration of MnTMPyP (2.5 µM) partially decreased fEPSP recovery post-hypoxia but not after OGD. In addition, post-ischemic LTP was impaired with MnTMPyP (2.5 µM) post-hypoxia but not post-OGD, suggesting an overall heightened sensitivity of hypoxia to MnTMPyP treatment.

In our electrophysiological studies, 25 μM MnTMPyp decreased baseline synaptic transmission over a 1h time-course. A sustained decrease in the fEPSP slope may be interpretated as a reduced amount of synaptic glutamate release [43]. A possible reason for this is altered vesicular trafficking leading to decreased AMPA receptor activity, which may account for this reduced fEPSP response [44]. This hypothesis is supported by our proteomics data, which highlight impaired vesicular trafficking mechanisms with MnTMPyP treatment compared to controls. In our studies, there was a significantly (*p* < 0.01) reduced Secretory Carrier Membrane Protein 3 (*SCAMP3*) gene expression as the protein with the most dramatic fold change difference. SCAMP3 is an abundant transmembrane protein found in endosomes and plays a central role in the intracellular trafficking of various transporters and receptors [45] and is a reported AMPA receptor-carrying vesicle [46]. Additionally, the observed downregulation of Neurogranin (*Nrgn)* may also impair glutamatergic currents via hypo-phosphorylation of NMDA receptors [47]. In addition, MnTMPyP reduced RAB7A signalling compared to controls. RAB proteins are small monomeric GTPases that are highly involved in the maintenance of neuronal vesicular trafficking, and RAB7 in particular is considered a marker for axonal retrograde transport [48] suggesting pre-synaptic effects. Voltage-dependent anion channel (VDAC) signalling was also reduced, which, due to its role in promoting apoptosis, may account for the observed neuroprotective effects of this compound [49]. There was also an impairment of LTP with MnTMPyP application. This result may be explained by impaired vesicular trafficking, since the expression of LTP is highly dependent on the increase of the probability of neurotransmitter release [50]. Additionally, our reported reduction in the microtubule-associated protein tau following MnTMPyP-treatment may impair LTP [51] via its effects on neuronal polarity and stability. The modulatory effects of ROS on LTP induction have been previously explored [15,52,53]. Klann et al. reported that application of 25 µM MnTMPyP significantly impairs LTP in vitro in the CA1 region of the hippocampus [54]. However, Arias-Cavieres et al. (2021) demonstrated that in vivo treatment with MnTMPyP rescued LTP deficits following intermittent long-term hypoxia treatments by modulating NMDAr subunit expression [55]. These results highlight that there may be differences in the outcome of MnTMPyP treatment depending on the concentration injected or time of application. As chronic MnTMPyP application appears to be protective, this suggests that it may act as a modulator for the production of receptor proteins rather than influence their location, conductance or number during ischemia and reperfusion. It is possible that this is an adaptive response in an attempt to overcome stress by minimizing the likely event that NMDARs become overactivated in response to the pro-oxidant environment. These in vivo experiments used MnTMPyP at a dose of 1 to 15 mg/kg i.p., and it will be important in the future to understand the biological significance of the concentrations used in vitro, namely 2.5 to 25 μM.

OGD is generally considered a more detrimental component of ischemia compared to hypoxia. The combination of both oxygen and glucose deprivation elicits a highly complex pathophysiology as neurons are particularly dependent on blood supply to sustain their oxygen and glucose demand [47]. Quantitative proteomics analysis by Datta et al. revealed that OGD caused a downregulation of proteins involved in protein metabolism and anti-oxidative response, while certain anti-apoptotic and anti-inflammatory, and most of the mitochondrial proteins, were increased in abundance [56]. Similarly, our protein enrichment analysis after OGD recovery showed upregulation of CAMKII, promoting glutamatergic plasticity and LTP [30], mitochondrial maintenance [31] and beta-tubulin, mediating microtubule transport of organelles and vesicles. The top canonical pathways additionally reflect a mitochondrial change (predicted upregulation of dysfunction after OGD; however, z-score is non-significant). Overall, this reflects our electrophysiology results, showing full recovery of fEPSP following 10 min OGD.

MnTMPyP treatment (25 µM) has previously been shown to be associated with a lack of recovery of fEPSPs following a 20 min OGD episode. This sustained reduction in fEPSP might be interpretated as neuroprotective through reduced glutamate release and subsequent decrease in the probability of post-ischemic excitotoxicity. Shahraki and Stone (2004) reported that superoxides can prevent the protective effect of adenosine presynapically and that this antagonism of adenosine responses may contribute to the injurious effects of ROS [57]. In addition, the application of DPCPX, DCKA and AP5 have been previously reported to reverse the protective effects of MnTMPyP on synaptic transmission post-ischemia [20]. It has been reported that superoxide is produced by NMDA receptors in an attempt to achieve a state of NMDA receptor hypofunction in order to limit damaging Ca^2+^ influx [58].

In these electrophysiological experiments, a shorter episode of OGD was used in order to be able to obtain a recovery of fEPSP and induce LTP. Application of MnTMPyP (25 µM) prior to a 10 min OGD induced a similar lack of fEPSP recovery to the 15 min OGD previously reported [20], an effect that was not seen with 2.5 µM. Proteomic analysis of 20 min OGD without MnTMPyP shows upregulation of synaptic plasticity and LTP. However, proteomic analysis following 25 µM MnTMPyP showed a reduction of signatures related to cell division with downregulation of Transforming Growth Factor Beta (TGFβ) [59] and increased expression of the Cyclin-dependent Kinase Inhibitor CDKN1B. Similarly, Yan et al. (2010) demonstrated that short bouts of OGD (1–2 h) inhibited the release of regulatory factors from astrocytes that promote neuronal differentiation and division [60]. While 20 min OGD without MnTMPyP induced a predicted upregulation of mitochondrial dysfunction, in agreement with its hypothesised neuroprotective function, MnTMPyP administration downregulates the pro-apoptotic Myc due to its regulatory role in *Bcl-2* [39]. Similarly, the top canonical pathways highlight Granzyme A (undetermined directionality), an important enzyme that mediates mitochondrial apoptosis [61]. This adds rigour to the neuroprotective properties of this compound, as mitochondrial dysfunction can be detrimental to neuronal function due to the excessive overproduction of ROS, leading to a loss of synaptic integrity and cell death [6].

When interpretating these results, it is important to take into account the age of the animals used in these experiments. For both electrophysiological and proteomic experiments, hippocampal slices from young rats (P21–28) were used. Many important stages of brain development in rodents occur in the early postnatal period. Over the first two weeks of development, synaptogenesis and myelination begin and neurogenesis is complete in the cortex but incomplete in the hippocampus and cerebellum [62,63]. Most of the pyramidal cells in the hippocampus are developed prenatally; however, only approximately 15% of granule cells are generated in the dentate gyrus at birth [64]. The most significant period of synaptogenesis in rats occurs up to P21 [65] and the majority of synaptic reorganisation has been shown to be established by postnatal day 10 [66,67,68,69]. In turn, it has been shown that acute slices usually have minimally altered synaptic connection patterns relative to the in vivo patterns at the time of harvest, adding rigour to the comparability of these models [6,70]. However, when interpretating these results, any aberrant impact on the data due to the age of the tissue and some active neurodevelopment at this stage must be taken into account.

It has previously been reported that MnTMPyP treatment (25 µM) caused a lack of recovery of fEPSPs following hypoxia (20 min) that has not been previously described [20]. This sustained reduction in fEPSP might also be interpreted as neuroprotective through a decrease in the probability of post-ischemic excitotoxicity. Pre-synaptic A_1_ receptor antagonism with DPCPX partially reversed this MnTMPyP effect, allowing the fEPSP to recover post-hypoxia, which may be interpreted as a negative effect on neuronal viability. Similarly, in these experiments, a lower concentration of MnTMPyP (2.5 µM) caused a significant lack of recovery post-hypoxia, but to a lesser extent.

Our proteomic data demonstrated that hypoxia treatment induced an increase in synaptogenesis compared to controls, perhaps to overcome the neuronal stress which may account for the full fEPSP recovery post-hypoxia. A study by Kovalenko et al. (2006) reported an increase in post-synaptic densities (PSDs) and multiple spine boutons (MSBs) post-hypoxia in the CA1, suggesting the induction of synaptogenesis mechanisms [71]. Interestingly, both MnTMPyP and hypoxia treatment significantly reduced synaptogenesis mechanisms, which may account for the reduced fEPSP recovery post-hypoxia. In particular, actin signalling, a cytoskeletal component key to neurogenesis and plasticity, comprised a central node of our most significant network and was dramatically upregulated following hypoxia in contrast to the administration of MnTMPyP. A study by Peinado et al. (2014) detected a similar modification of actin nitration patterns, suggesting that there may be a hypoxia-derived impairment of cell structure [72]. This modulation of actin during hypoxia may also account for the impairment of post-ischemic LTP with MnTMPyP treatment seen in our electrophysiological data, as actin is known to play a central role in the biogenesis, transport and anchoring of synaptic vesicles, an essential component in the expression of LTP [73]. In addition, mitochondrial dysfunction was predicted to be significantly reduced in MnTMPyP and hypoxia compared to the hypoxia-only groups, suggesting a neuroprotective effect. Interestingly, edaravone, a free radical scavenger and antioxidant, has previously been reported to have a neuroprotective effect, reducing apoptosis and stimulating the BDNF/TrkB pathway, crucial in neural development, neurogenesis and synaptic plasticity [74,75]. Preliminary data in our laboratory indicated that prior treatment of organotypic hippocampal slices exposed to a 1 h OGD insult with edaravone showed improved cell viability up to 24 h.

Administration of a hypoxic insult to CA1 hippocampal slices elicits a distinct proteomic signature compared to oxygen and glucose deprivation. At the centre of the most enriched network for hypoxic recovery is marked upregulation of actin, a key component of synaptogenesis, in addition to upregulation of Heat Shock protein Hsp90, indicative of recovery from cell stress. It has been previously reported that hypoxia, via various mechanisms, can modulate actin cytoskeleton dynamics [76]. The enriched network generated from OGD does not reflect as marked an upregulation of actin, but instead shows increased expression of calmodulin-associated protein CAMKII and microtubule signalling. CAMKII also plays crucial roles in mitochondrial maintenance [31]. Furthermore, actin signalling does not appear amongst the most significant pathways in OGD, with mitochondrial dysfunction amongst the top three signatures. OGD is reported in the literature to impact mitochondrial morphology and protein expression in neuronal cultures [77]. Thus, it seems hypoxia is characterised by the recovery of synaptogenesis via actin-mediated signalling, whilst OGD recovery emphasises mitochondrial functioning.

MnTMPyP treatment (2.5 µM, 25 µM) impaired recovery of fEPSPs following hypoxia (20 min) and following a 20 min OGD episode; however, OGD was not sensitive to MnTMPyP at 2.5 µM. This elevated sensitivity of hypoxia-treated slices to MnTMPyP may be due to this increased emphasis on actin regeneration. Indeed, the literature supports the role of ROS in promoting actin monomer incorporation into filaments, with MnTMPyP administration shown to virtually abolish actin polymerisation due to reduced available ROS levels [78]. This is reflected in the reduced upregulation of actin as a central node in network analysis of MnTMPyP administration to hypoxic slices. The mechanisms through which OGD alone affects synaptogenesis concerns mitochondrial dysfunction. Mitochondria play a key role in bioenergetics and cell death, and thus are central in the appropriate regulation of synaptic budding and pruning [79]. Amongst our most significant implicated canonical pathways of MnTMPyP administration in the normoxia conditions above is a predicted reduction in mitochondrial dysfunction. This is in agreement with the role of the drug in reducing ROS, as this may damage mitochondria and impact respiration [80]. Indeed, the central node of the most enriched pathway in OGD following MnTMPyP treatment concerned downregulation of TGFβ, promoting mitochondrial respiration [38]. Hence, MnTMPyP may partially compensate for mitochondrial dysfunction underlying OGD-mediated synaptogenesis and fEPSP impairments.

Our proteomic analysis features upregulation of Heat Shock Proteins (HSPs) in the hypoxia-treated group that was rescued by MnTMPyP application. HSPs are molecular chaperones produced in response to oxidative stress and an increase in ROS production [81]. Zatsepina et al. (2021) demonstrated that HSPs are critically involved in the processes of protein synthesis and synaptic receptor trafficking necessary for the maintenance of synapses and the expression of LTP [82]. In addition, Gerges et al. reported that the Hsp90 family of proteins are specifically involved in the trafficking of AMPAr and neurotransmitter release [83], both of which are essential for LTP maintenance. The electrophysiological results support this notion where full post-ischemic LTP is achieved. However, an impaired expression of post-hypoxic LTP is observed specifically when MnTMPyP is applied. The reduction of HSP expression with MnTMPyP treatment (reduced ROS) may account for this change due to the central role of HSPs in the generation of LTP during episodes of oxidative stress. As our proteomic analysis reported no HSP signature changes in OGD, this may explain why post-hypoxia LTP was impaired and full LTP was achieved post-OGD with MnTMPyP treatment. In addition, the enriched network generated from OGD reflects an increased expression of CAMKII not seen in hypoxia. This may also partially explain the OGD-specific expression of post-ischemic LTP with MnTMPyP. Early LTP involves CaMKII-dependent phosphorylation of the AMPAr subunit, GluR1, which increases channel conductance of AMPARs. In addition, CAMKII phosphorylates stargazin, which allows extrasynaptic AMPARs to bind to postsynaptic density protein 95 (PSD95), thereby anchoring more AMPARs at the synapse and facilitating LTP expression [30].

## 5. Conclusions

Our electrophysiological and proteomic results have demonstrated a novel role for the antioxidant MnTMPyP in modulating hippocampal (CA1) synaptic plasticity after acute hypoxia and OGD. MnTMPyP was shown to cause an impairment of post-ischemic LTP, suggesting a heightened sensitivity of hypoxia to MnTMPyP treatment compared to OGD. This difference in sensitivity may be due to the increased expression of CAMKII seen in OGD but not hypoxia. In addition, the reduction of HSP expression and actin signalling in hypoxia with MnTMPyP treatment may account for this change due to the central role of HSPs and actin in vesicular trafficking mechanisms and the generation of LTP during episodes of oxidative stress. As our proteomic analysis reported no HSP or actin signature changes in OGD, this may explain why LTP was impaired post-hypoxia but not post-OGD with MnTMPyP treatment. Elucidating the exact mechanistic action of MnTMPyP and other antioxidants on synaptic plasticity during episodes of ischemic stress will be important. Finally, further validation of the role of these proteins will be required and must be considered a limitation of this study.

## Figures and Tables

**Figure 1 antioxidants-12-00792-f001:**
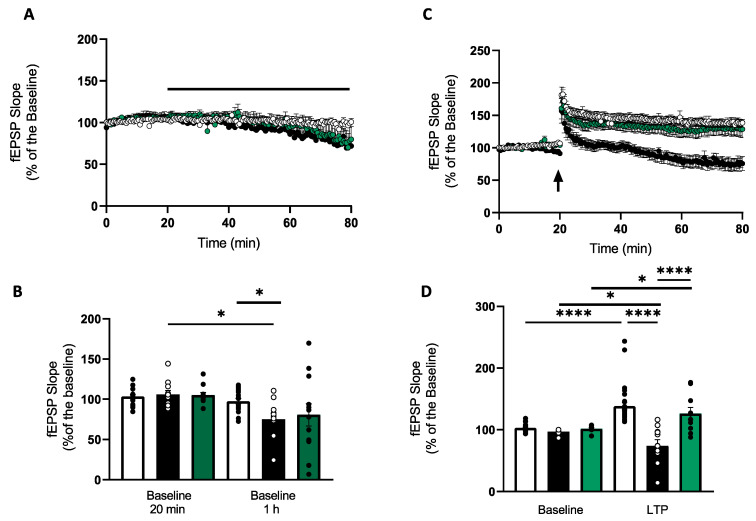
The effects of MnTMPyP on synaptic transmission and plasticity. (**A**) Time-course showing changes in CA1 fEPSP baseline slope over 1 h for controls (white circles), 2.5 µM MnTMPyP (green circles) and 25 µM MnTMPyP (black circles). In controls and 2.5 µM MnTMPyP-treated groups (2.5 µM MnTMPyP; 20 min: 105.3 ± 3.35%; 1 h: 81.9 ± 14.1%; *p* > 0.05; *n* = 11), there was no significant change in fEPSP slope over a 1 h time-course (controls; 20 min: 103 ± 2.5%; 1 h: 98 ± 3.3%; *p* > 0.05; *n* = 19). However, in the presence of 25 µM MnTMPyP, there was a significant decrease in fEPSP slope over 1 h (baseline 20 min: 105.9 ± 5.2%; baseline 1 h: 76.3 ± 7.5%; *n* = 11; *p* < 0.05). (**B**) Bar chart summarising the time-course data in A. White bars represent controls (baseline at 20 min and 1 h timepoints) at baseline (5 min on baseline transmission taken before the addition of MnTMPyP) and last 5 min of 1 h drug incubation. Black bars represent 25 µM MnTMPyP at 20 min and 1 h timepoints. Green bars represent 2.5 µM MnTMPyP-treated group at the same 20 min and 1 h time points. (**C**) Time-course of fEPSP slope before and after HFS from control hippocampal slices (white circles), hippocampal slices pretreated with 25 µM MnTMPyP (dark circles) and hippocampal slices pretreated with 2.5 µM MnTMPyP (green circles). Baseline is normalised to the first 20 min of control fEPSP. HFS stimulation in control slices (white circles) gave rise to a robust synaptic potentiation (135.0 ± 6.1%, *n* = 27) after 60 min. MnTMPyP (2.5 µM, green circles) did not impair synaptic plasticity (121.0 ± 9.2, *n* = 9). However, high concentrations of the superoxide dismutase mimetic (25 µM, black circles) impaired synaptic potentiation and significantly reduced baseline synaptic transmission 60 min after the HFS (74.3 ± 9.3%, *n* = 11). (**D**) Bar charts summarising the data in C. Control LTP (white bars), 25 µM MnTMPyP (black bars) and 2.5 µM MnTMPyP (green bars). Baseline values were taken 5 min before HFS induction. Average fEPSP slope was taken at 60 min after HFS. Arrow indicates timing of HFS. All data are expressed as mean ± SEM. * *p*  <  0.05, **** *p*  <  0.0001.

**Figure 2 antioxidants-12-00792-f002:**
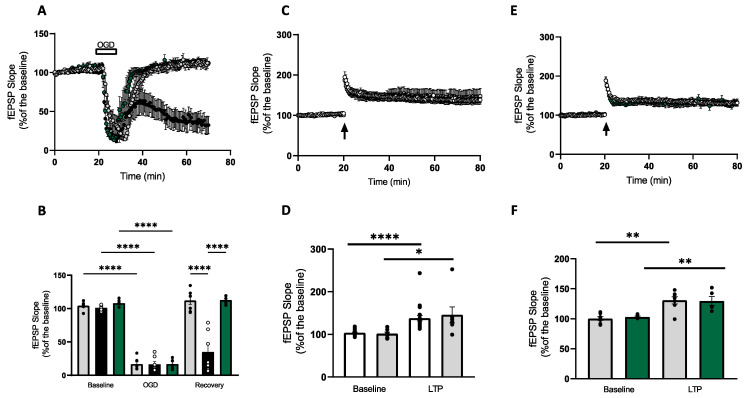
MnTMPyP does not affect synaptic transmission during OGD and post-ischemic LTP. (**A**) Timeline showing changes in CA1 fEPSP slope following a 10 min OGD episode for control (grey circles), MnTMPyP (25 µM, black circles) and MnTMPyP (2.5 µM, green circles). In the CA1, OGD gave rise to a maximum depression of fEPSP to 48.0 ± 10.0% of control and a recovery to 99.5 ± 10.1% (*n* = 5). In slices pretreated with 25 µM MnTMPyP, the OGD-induced depression of fEPSP was 16.4 ± 4.2%, with a recovery of 35.0 ± 11.4, *n* = 7 at 40 min post-OGD. OGD slices pretreated with 2.5 µM MnTMPyP had similar depression of fEPSP but modulated the recovery after the ischemic insult (17.0 ± 3.4% and 113.0 ± 3.0%, respectively, *n* = 5, at 40 min post-OGD). (**B**) Bar charts summarising the data in A. Grey bars represent controls, black bars represent 25µM MnTMPyP and green bars represent 2.5 µM MnTMPyP-treated group. (**C**) Time-course showing control LTP (white circles) and LTP post-10 min OGD (grey circles). HFS induced a stable LTP in control slices lasting 60 min (135.0 ± 6.1%, *n* = 27) with no difference compared to LTP induced in slices pre-exposed to OGD (131.0 ± 6.0%, *n* = 7). (**D**) Bar charts summarising the data in C. White bars control and grey bars post-OGD. (**E**) Time-course of fEPSP slope before and after HFS from control slices (grey circles) and 2.5 µM MnTMPyP-pretreated slices (green circles). HFS was induced 40 min after the ischemic event. In control slices, the stimulation gave rise to a stable synaptic potentiation after 60 min (131.0 ± 6.0%, *n* = 7). In slices pretreated with 2.5 µM MnTMPyP, the HFS induced a robust LTP with no significant difference compared to post-ischemia control LTP (129.5 ± 7.5%, *n* = 4). (**F**) Bar charts summarising the data in E. Grey bars represent controls and green bars represent post-2.5 µM MnTMPyP. In LTP experiments, baseline values were taken 5 min before HFS induction, LTP values were taken at 55 min after HFS. Arrow indicates timing of HFS. All data are expressed as mean ± SEM. * *p*  <  0.05, ** *p*  <  0.01, **** *p*  <  0.0001.

**Figure 3 antioxidants-12-00792-f003:**
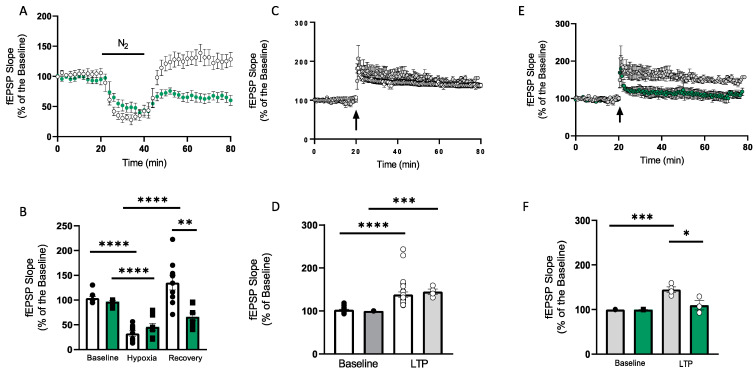
Application of MnTMPyP modulates synaptic transmission and plasticity post-hypoxia. (**A**) Time-course showing changes in CA1 fEPSP slope following a 20 min hypoxic episode for control (white circles) and MnTMPyP (2.5 µM, green circles). In the CA1, hypoxia gave rise to a maximum depression of fEPSP slope to 32.45 ± 6.8% and full recovery (135.41 ± 12%) upon reoxygenation (*n* = 9). In slices pretreated with 2.5 µM MnTMPyP, the hypoxia-induced fEPSP slope depression was 45.8 ± 7.4% (*n* = 8), with partial recovery upon reoxygenation (66.24 ± 8%; *n* = 8). (**B**) Bar chart summarising the time-course data in A. White bars represent controls, green bars represent MnTMPyP-treated group. (**C**) Time-course of fEPSP slope before and after HFS from control hippocampal slices (white circles) and from hippocampal slices exposed to 20 min hypoxia (grey circles). Time-course showing control LTP (white circles; *n* = 27) and LTP post-20 min hypoxia (grey circles; *n* = 4). HFS induced a stable LTP in control slices lasting 60 min (135.0 ± 6.1%, *n* = 27) with no difference compared to LTP induced in slices pre-exposed to hypoxia (144.87 ± 5.67%; *n* = 4). (**D**) Bar charts summarising the data in C. Control (white bars), post-hypoxia (grey bars). Baseline values were taken 5 min before the induction of LTP. LTP fEPSP slope was taken at 60 min post-HFS (white and grey bars). (**E**) In LTP slices pre-treated with hypoxia, HFS was induced 40 min after the 20 min hypoxic episode. Here, the stimulation gave rise to a stable synaptic potentiation lasting for 60 min (144.87 ± 5.67%; *n* = 4). Application of 2.5 µM MnTMPyP (green circles) significantly impaired LTP post-hypoxia (113.4 ± 11.9%; *n* = 4). (**F**) Bar chart summarising the data in E. Control (grey bars), post-hypoxia (green bars). Arrow in C,E indicates timing of HFS. All data are expressed as mean ± SEM. * *p*  <  0.05, ** *p*  <  0.01, ** *p*  <  0.001 **** *p*  < 0.0001.

**Figure 4 antioxidants-12-00792-f004:**
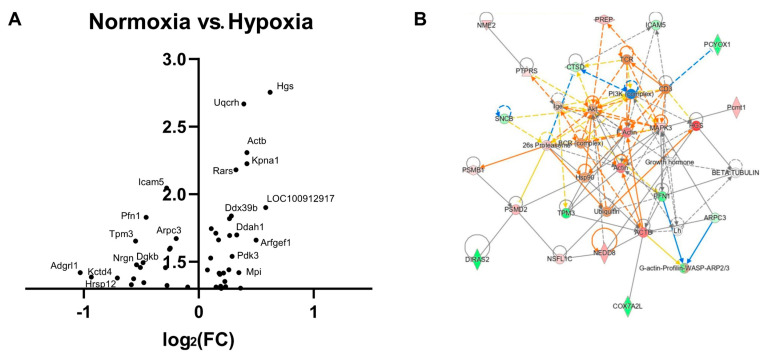
Actin signalling and phagocytosis are upregulated in post-hypoxia recovery. (**A**) A volcano plot visualisation of proteomic changes shows dramatically altered expression of targets implicated in synapse formation (Adgrl1) and potassium channel tetramerization (Kctd4), with endosomal trafficking (Hgs) and mitochondrial respiration (Uqcrh) amongst the most significant hits. (**B**) Ingenuity network enrichment analysis (*p* = 1.00 × 10^−52^) highlights actin as a highly upregulated central node (Actin, ACTB), in addition to upregulation of the BCR complex in antigen presentation and phagocytosis.

**Figure 5 antioxidants-12-00792-f005:**
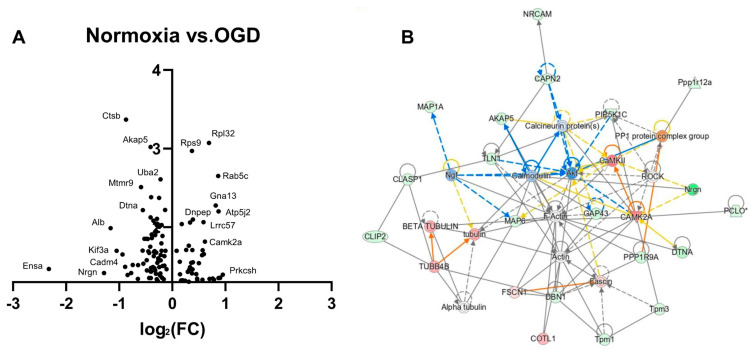
Post-OGD recovery reveals upregulation of actin signalling and mitochondrial dysfunction. (**A**) A volcano plot of proteomic changes shows dramatically changed expression of targets involved in potassium channel regulation (Ensa) and LTP (Nrgn), while the most significantly associated targets are implicated in lysosome function (Ctsb) and dendritic endosomal trafficking (AKAP5). (**B**) Ingenuity network enrichment analysis (*p* = 1.00 × 10^−48^) reveals actin to be a central node of the network analysis, in addition to targets regulating synaptic plasticity (CAMKII) and microtubule-mediated transport (beta-tubulin).

**Figure 6 antioxidants-12-00792-f006:**
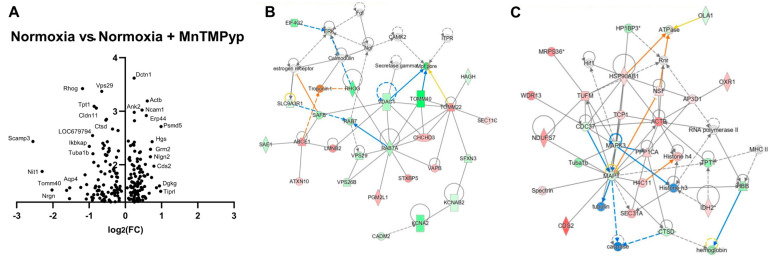
MnTMPyP applied in normoxia conditions modulates aspects of synaptogenesis, cellular stress and mitochondrial respiration. (**A**) Volcano plot showing significantly changed (*p* < 0.05) proteins across the two groups. The most dramatically altered proteins (log2(Fold Change) > 2) are implicated in endocytosis (Scamp3), antioxidant activity (Nit1), synaptic development and plasticity (Nrgn). (**B**,**C**) Network enrichment analysis shows two networks of equal significance (*p* = 1.00 × 10^−48^), one with downregulation of proteins implicated in apoptosis, endosomal trafficking and mitochondrial function (**B**), and the second centred on the downregulation of the protein tau, essential for maintaining neuronal polarity (**C**).

**Figure 7 antioxidants-12-00792-f007:**
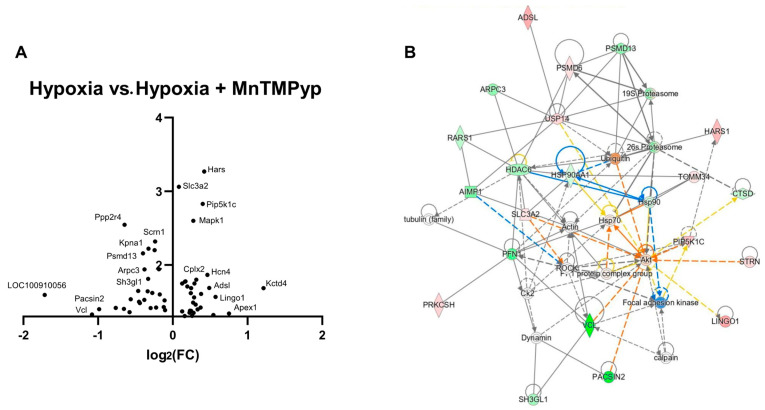
Application of MnTMPyP downregulates synaptogenesis signalling following hypoxic insult. (**A**) A volcano plot shows major upregulation of potassium-channel-associated Kctd4 and downregulation of LOC100910056, also knows as Bltp3b, involved in endosomal vesicle transport. (**B**) The most enriched network (*p* = 1.00 × 10^−52^) is centred around actin, implicated in maintaining neuronal structure and integrity, in addition to numerous heat shock proteins (HSPAA1, HSP70; downregulated, HSP90; upregulated) responsible for cellular stress response.

**Figure 8 antioxidants-12-00792-f008:**
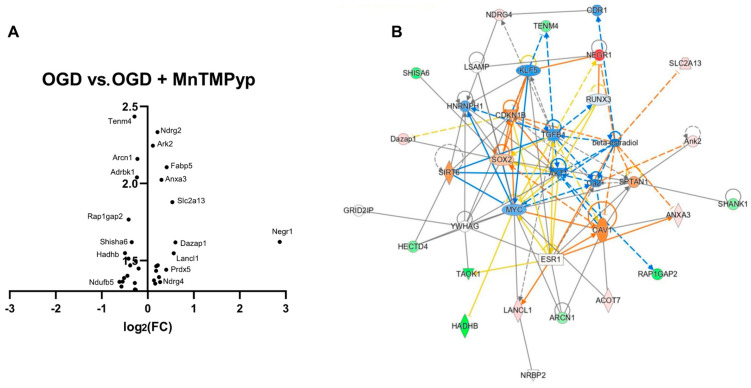
Application of MnTMPyP modulates synaptogenesis and mitochondrial oxidative phosphorylation following OGD. (**A**) Volcano plot of differentially expressed targets shows the most dramatic upregulation (log2(Fold Change) > 2) of Negr1, responsible for axon growth. The most significant hits include downregulation of Tenm4 (axonal myelination) and upregulation of Ndrg2 (neurite outgrowth). (**B**) The most enriched network (*p* = 1.00 × 10^−42^) is centred around downregulation of TGFB1, a growth factor with roles in proliferation, differentiation and growth, in addition to actin dynamics and mitochondrial activity. Other central nodes include CDKN1B, a regulator of cell cycle, and downregulation of beta-oestradiol, with reported neuroprotective functions.

**Figure 9 antioxidants-12-00792-f009:**
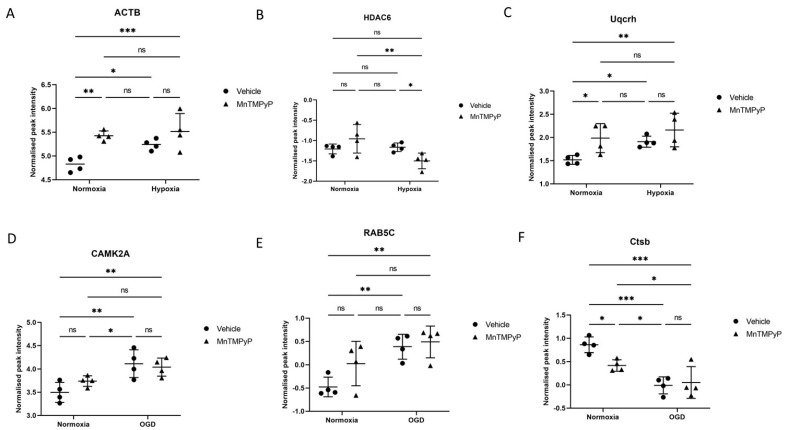
Two-way ANOVA analysis across hypoxia-treated and OGD-treated CA1 region replicates previously highlighted themes of plasticity, cell stress and mitochondrial change. Comparison of key targets across normoxia, hypoxia, normoxia + MnTMPyP and hypoxia + MnTMPyP groups via two-way ANOVA. Scatter plots highlight (**A**) Actin (ACTB), (**B**) Histone deacetylase 6 (HDAC6) and (**C**) (Uqcrh), implicated in synaptogenesis and neuronal structural integrity, cell stress response and mitochondrial electron transport chain, respectively. Comparison of normoxia, OGD, normoxia + MnTMPyP and OGD + MnTMPyP via two-way ANOVA yields scatter plots highlighting (**D**) CAMK2A, (**E**) RAB5C and (**F**) Ctsb, implicated in glutamatergic synaptic plasticity and mitochondrial maintenance, dendritic branching and endosomal trafficking, and apoptosis and neuroinflammation, respectively. * *p*  <  0.05, ** *p*  <  0.01, *** *p*  <  0.001.

**Table 1 antioxidants-12-00792-t001:** Ingenuity canonical pathways post-hypoxia. Ingenuity pathway analysis, ranked by -log (*p*-value) highlights actin upregulation amongst the most significant changes (*p* = 5.3 × 10^−5^, z = 0.4), in addition to phagocytosis signalling (*p* = 5.2 × 10^−7^, z = 1.3), compared to normoxia-treated slices.

Ingenuity Canonical Pathways	−Log (*p*-Value)	z-Score	Ratio	Molecules
Fcγ Receptor-mediated Phagocytosis in Macrophages and Monocytes	6.28	1.34	0.053	ACTB, ARPC3, DGKB, MAPK3, TLN2
Integrin Signalling	4.55	0.45	0.024	ACTB, ARPC3, MAPK3, PFN1, TLN2
Actin Cytoskeleton Signalling	4.26	0.45	0.021	ACTB, ARPC3, MAPK3, PFN1, TLN2
Inhibition of ARE-Mediated mRNA Degradation Pathway	3.79		0.025	MAPK3, PPP2R1A, PSMB1, PSMD2
Remodelling of Epithelial Adherens Junctions	3.68		0.044	ACTB, ARPC3, HGS
BAG2 Signalling Pathway	3.41		0.036	MAPK3, PSMB1, PSMD2
Clathrin-mediated Endocytosis Signalling	3.38		0.019	ACTB, ARPC3, HGS, PCYOX1
Ceramide Signalling	3.31		0.033	CTSD, MAPK3, PPP2R1A
Salvage Pathways of Pyrimidine Ribonucleotides	3.23		0.031	AK1, MAPK3, NME2
Pyrimidine Deoxyribonucleotides De Novo Biosynthesis I	3.16		0.087	AK1, NME2
Regulation of Actin-based Motility by Rho	3.01		0.026	ACTB, ARPC3, PFN1
RHOA Signalling	2.92		0.024	ACTB, ARPC3, PFN1
Huntington’s Disease Signalling	2.88		0.014	CTSD, MAPK3, PSMB1, PSMD2
Reelin Signalling in Neurons	2.78		0.022	ARPC3, MAPK3, PDK3
D-mannose Degradation	2.77		1	MPI
Pyrimidine Ribonucleotides Interconversion	2.72		0.053	AK1, NME2
Pyrimidine Ribonucleotides De Novo Biosynthesis	2.66		0.049	AK1, NME2
Aryl Hydrocarbon Receptor Signalling	2.61		0.019	CTSD, MAPK3, NEDD8
Thyroid Hormone Biosynthesis	2.47		0.5	CTSD
FAT10 Signalling Pathway	2.39		0.036	PSMB1, PSMD2
Production of Nitric Oxide and Reactive Oxygen Species in Macrophages	2.38		0.016	MAPK3, PCYOX1, PPP2R1A
ILK Signalling	2.32		0.015	ACTB, MAPK3, PPP2R1A
ERK/MAPK Signalling	2.24		0.014	MAPK3, PPP2R1A, TLN2
Agrin Interactions at Neuromuscular Junction	2.21		0.029	ACTB, MAPK3
Caveolar-mediated Endocytosis Signalling	2.14		0.027	ACTB, COPB2
AMPK Signalling	2.1		0.012	ACTB, AK1, PPP2R1A

**Table 2 antioxidants-12-00792-t002:** Ingenuity canonical pathway analysis post-OGD. Significant ingenuity pathway analysis, ranked by −log (*p*-value), shows synaptic plasticity (PKA; *p* = 5.3 × 10^−5^) and OGD-associated transcript degradation (*p* = 5.8 × 10^−5^), in addition to mitochondrial dysfunction (*p* = 9.5 × 10^−5^) amongst the top pathways, compared to normoxia-treated slices.

Ingenuity Canonical Pathways	−Log (*p*-Value)	z-Score	Ratio	Molecules
Protein Kinase A Signalling	4.27	0	0.022	AKAP5, CAMK2A, GNA13, GNB1, PRKAR2B, PTK2B, PTPN23, YWHAG, YWHAH
Inhibition of ARE-Mediated mRNA Degradation Pathway	4.23		0.037	LOC100360846/Psmb6, PRKAR2B, PSMA4, PSMA7, YWHAG, YWHAH
Mitochondrial Dysfunction	4.02	0.71	0.023	CAMK2A, CAPN2, MAOA, MCU, PARK7, PRKAR2B, SOD2, UQCRQ
Calcium Signalling	3.5		0.027	AKAP5, CAMK2A, MCU, PRKAR2B, Tpm1, Tpm3
BAG2 Signalling Pathway	3.38		0.048	CTSB, LOC100360846/Psmb6, PSMA4, PSMA7
Glycine Betaine Degradation	3.12		0.2	SHMT2, SRR
FAT10 Signalling Pathway	2.79		0.054	LOC100360846/Psmb6, PSMA4, PSMA7
Clathrin-mediated Endocytosis Signalling	2.75		0.024	ALB, CLU, PIP5K1C, RAB5C, SH3GL1
RHOA Signalling	2.75	0	0.032	GNA13, PIP5K1C, Ppp1r12a, PTK2B
ERK/MAPK Signalling	2.69		0.023	PRKAR2B, PTK2B, TLN1, YWHAG, YWHAH
Ephrin B Signalling	2.47		0.042	GNA13, GNB1, NCK2
ERK5 Signalling	2.44		0.041	GNA13, YWHAG, YWHAH
Chemokine Signalling	2.33		0.037	CAMK2A, Ppp1r12a, PTK2B
Signalling by Rho Family GTPases	2.29		0.019	GNA13, GNB1, PIP5K1C, PTK2B, STMN1
Crosstalk between Dendritic Cells and Natural Killer Cells	2.19		0.033	CAMK2A, FSCN1, TLN1
Huntington’s Disease Signalling	2.18		0.018	CAPN2, GNB1, LOC100360846/Psmb6, PSMA4, PSMA7
Cardiac β-adrenergic Signalling	2.17		0.022	AKAP5, GNA13, GNB1, PRKAR2B
Fcγ Receptor-mediated Phagocytosis in Macrophages and Monocytes	2.15		0.032	NCK2, PTK2B, TLN1
Oestrogen Receptor Signalling	2.14	0.82	0.015	GNA13, GNB1, MCU, Ppp1r12a, PRKAR2B, SOD2
IL-1 Signalling	2.12		0.031	GNA13, GNB1, PRKAR2B
Glycine Biosynthesis I	2.08		0.5	SHMT2
GNRH Signalling	2.08	0	0.021	CAMK2A, GNB1, PRKAR2B, PTK2B
Xenobiotic Metabolism PXR Signalling Pathway	2.07	−1	0.021	CAMK2A, GSTM5, MAOA, PRKAR2B
IGF-1 Signalling	2.02		0.029	PRKAR2B, YWHAG, YWHAH
Paxillin Signalling	2		0.028	NCK2, PTK2B, TLN1

**Table 3 antioxidants-12-00792-t003:** Ingenuity pathway analysis of normoxia-treated CA1 slices following MnTMPyP administration, compared to vehicle-treated normoxia. Ingenuity-generated associated canonical pathways, ranked by −log (*p*-value), with the top three pathways centred on synaptogenesis signalling (*p* = 6.3 × 10^−8^), cellular stress (BAG2) (*p* = 1.5 × 10^−7^) and mitochondrial function (*p* = 1.2 × 10^−6^).

Ingenuity Canonical Pathways	−Log (*p*-Value)	z-Score	Ratio	Molecules
Synaptogenesis Signalling Pathway	7.2	0.53	0.044	ARHGEF7, ARPC3, ARPC4, GRIA1, GRM2, MAPK1, MAPK3, MAPT, NAPB, NLGN2, NRXN1, NSF, PRKCE, STXBP5
BAG2 Signalling Pathway	6.81	0	0.095	CTSB, MAPK1, MAPK3, MAPT, PSMA3, PSMB4, PSMC5, PSMD5
Mitochondrial Dysfunction	5.91	−1.39	0.038	ATP1A2, ATP1B3, Cox7a2/Cox7a2l2, COX7A2L, CYB5A, IDH2, MAOA, MAOB, MAPT, NDUFS7, TOMM22, TOMM40, VDAC1
Integrin Signalling	5.52	−0.33	0.047	ACTB, ARF6, ARHGEF7, ARPC3, ARPC4, GIT1, MAPK1, MAPK3, PFN1, RHOG
Fcγ Receptor-mediated Phagocytosis in Macrophages and Monocytes	5.3	1.13	0.075	ACTB, ARF6, ARPC3, ARPC4, MAPK1, MAPK3, PRKCE
Huntington’s Disease Signalling	5.21		0.039	CTSD, DCTN1, MAPK1, MAPK3, NAPB, NSF, PRKCE, PSMA3, PSMB4, PSMC5, PSMD5
Remodelling of Epithelial Adherens Junctions	5.04	0.45	0.088	ACTB, ARF6, ARPC3, ARPC4, HGS, RAB7A
Clathrin-mediated Endocytosis Signalling	4.72		0.043	ACTB, ALB, AP3D1, ARF6, ARPC3, ARPC4, CLU, HGS, RAB7A
Sirtuin Signalling Pathway	4.3	0.38	0.034	H4C11, IDH2, MAP1LC3A, MAPK1, MAPK3, NDUFS7, PGAM1, TOMM22, TOMM40, VDAC1
Actin Cytoskeleton Signalling	4.18	0.71	0.037	ACTB, ARHGEF7, ARPC3, ARPC4, CYFIP1, GIT1, MAPK1, MAPK3, PFN1

**Table 4 antioxidants-12-00792-t004:** Ingenuity pathway analysis of CA1 slices treated with hypoxia + MnTMPyP, compared to vehicle-treated hypoxia. Ingenuity-generated associated canonical pathways, ranked by −log (*p*-value), show the top canonical pathways concern themes such as apoptosis regulation (Sirtuin, *p* = 1.2 × 10^−8^), cell stress regulation (BAG2, *p* = 3.31 × 10^−5^) and mitochondria (*p* = 9.77 × 10^−5^), in addition to actin (*p* = 1.82 × 10^−4^).

Ingenuity Canonical Pathways	−Log (*p*-Value)	z-Score	Ratio	Molecules
Huntington’s Disease Signalling	8.05		0.032	CPLX2, CTSD, GNG3, HDAC6, MAPK1, NAPB, PSMD13, PSMD6, SDHA
Sirtuin Signalling Pathway	7.92	0.82	0.031	APEX1, MAPK1, NDUFA10, NDUFA9, PGAM1, SDHA, SOD2, TIMM9, TOMM34
Oestrogen Receptor Signalling	4.57		0.017	GNG3, HSP90AA1, MAPK1, NDUFA10, NDUFA9, SDHA, SOD2
BAG2 Signalling Pathway	4.48		0.048	HSP90AA1, MAPK1, PSMD13, PSMD6
Mitochondrial Dysfunction	4.01	−2.45	0.017	NDUFA10, NDUFA9, OPA1, SDHA, SOD2, TOMM34
Actin Cytoskeleton Signalling	3.74	−0.45	0.021	ARPC3, MAPK1, PFN1, PIP5K1C, VCL
Neutrophil Extracellular Trap Signalling Pathway	3.59	2.45	0.015	MAPK1, NDUFA10, NDUFA9, SDHA, TIMM9, TOMM34
Granzyme A Signalling	3.24		0.04	APEX1, NDUFA10, NDUFA9
Clathrin-mediated Endocytosis Signalling	2.98		0.019	AP3D1, ARPC3, PIP5K1C, SH3GL1
Integrin Signalling	2.94	−1	0.019	ARPC3, MAPK1, PFN1, VCL

**Table 5 antioxidants-12-00792-t005:** Ingenuity pathway analysis of CA1 slices administered with OGD and MnTMPyP, compared to vehicle-treated OGD. Ingenuity-generated associated canonical pathways, ranked by −log (*p*-value), reveal the top canonical pathways concern mitochondrial-mediated apoptosis (Granzyme A, *p* = 4.07 × 10^−3^), oxidative phosphorylation (*p* = 8.7 × 10^−3^) and actin mobility (*p* = 9.33 × 10^−3^).

Ingenuity Canonical Pathways	−Log (*p*-Value)	z-Score	Ratio	Molecules
Granzyme A Signalling	2.39		0.027	NDUFB5, NDUFB9
Oxidative Phosphorylation	2.06		0.018	NDUFB5, NDUFB9
Regulation of Actin-based Motility by Rho	2.03		0.017	CFL1, PFN1

## Data Availability

All data in the manuscript are available from the corresponding author.

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
