# Peer review of "An Electrophysiological and Proteomic Analysis of the Effects of the Superoxide Dismutase Mimetic, MnTMPyP, on Synaptic Signalling Post-Ischemia in Isolated Rat Hippocampal Slices"

_antioxidants, 2023, doi:10.3390/antiox12040792_

Round 1
Reviewer 1 Report
The authors have studied the effect of MnTMPyP on the synaptic transmission in the CA1 region of the hippocampal slices. Further, the authors performed the proteomic analysis on these slices. The manuscript is well written although the discussion section is very lengthy. The authors didnot discuss the drawback of the study. The whole manuscript looks like a bioinformatics analysis. The conclusions are also not supported by the result. My comments are provided below
1. The authors use only two dose of MnTMPyP. The authors didnot perform any dose response study using electrophysiology. technique to claim that 25uM is the highest dose. Secondly, the authors didnot discuss the biological significance of the so called high dose of MnTMPyP.
2. After the proteomic study, the authors didnot validated the data using western blot analysis. The authors should show the changes in gene expression using western blot atleastfor some important genes.
3. From the gene signature studies (bioinformatic analysis), the authors claimed that the drug- MnTMPyP has worked on mitochondria, actin and synaptogenesis. This is the main drawback of the study. The authors should the effect of MnTMPyP on these process using other biochemical methods.
4. The other drawback of the study is lack of in vivo study to show the effect of MnTMPyP. The authors should discuss this point in the manuscript.
5. The authors have used five treatment groups in the proteomic analysis but the presentation was done for two groups at one time. Also the stastistical test was student t test. It will be better if the authors combined all the groups in one figure and use some strong statistical test.
Author Response
Referee 1
The authors have studied the effect of MnTMPyP on the synaptic transmission in the CA1 region of the hippocampal slices. Further, the authors performed the proteomic analysis on these slices. The manuscript is well written although the discussion section is very lengthy. The authors did not discuss the drawback of the study. The whole manuscript looks like a bioinformatics analysis. The conclusions are also not supported by the result.
My comments are provided below
1.The authors use only two dose of MnTMPyP. The authors did not perform any dose response study using electrophysiology technique to claim that 25uM is the highest dose. Secondly, the authors did not discuss the biological significance of the so called high dose of MnTMPyP.
The referee is correct and we cannot claim that 25 mM MnTMPyp is the highest dose or has the maximal effect. We have taken out reference to concentration dependent effects (Page 17 and 19). We use the term higher concentration in the manuscript (as opposed the 2.5 mM concentration). We have previously published electrophysiological data with MnTMPyP at 25 mM (Moreton et al., 2022), where we demonstrated a complete lack of fEPSP recovery post-ischemia. 25 mM has been previously used in the literature in vitro (Klann 1998) where it has been shown to significantly impair LTP. This manuscript also shows impaired LTP at 25 mM. We decided to use a 10 times lower concentration in some of our experiments to see if the non-recovery of synaptic transmission post hypoxia and impaired LTP could be attenuated. This indeed was the case in this study. A 2.5 mM to 25 mM concentration also seems to correlate with the reported in vivo doses used to date, namely 1 to 15mg/kg, i.p (ref 18, 19) and which also have modulatory effects on synaptic plasticity and neuroprotection. We have now inserted reference to this information in our discussion on page 22.
- After the proteomic study, the authors did not validated the data using western blot analysis. The authors should show the changes in gene expression using western blot at least for some important genes.
We are hoping to do western blot analysis on some of these important proteins in the near future but are not in this position at the moment. We hope the electrophysiological and proteomic analysis is sufficient at the moment, as we feel it is important for it to be published. In the Conclusions on Page 21 we now include a sentence outlining that a drawback of this study is the lack of validation of the proteomics data.
- From the gene signature studies (bioinformatic analysis), the authors claimed that the drug- MnTMPyP has worked on mitochondria, actin and synaptogenesis. This is the main drawback of the study. The authors should the effect of MnTMPyP on these process using other biochemical methods.
As mentioned above the Referee is absolutely correct and this is a drawback of the study. We do intend to investigate MnTMPyP using other biochemical methodologies but at present we would like to publish the data and make the raw data available to all the community. The final sentence of the conclusion outlines this limitation of the study.
- The other drawback of the study is lack of in vivo study to show the effect of MnTMPyP. The authors should discuss this point in the manuscript.
In the introduction we do report two in vivo studies by Khuu et al. (ref 18) and Arias-Cavieres et al. (ref 19) where they demonstrated that in vivo treatment (via intraperitoneal injection) with MnTMPyP rescue hippocampal synaptic plasticity (LTP) and ameliorates neurogenesis in an intermittent hypoxia murine model. In the Discussion we also refer to the Arias-Cavieres experiments in vivo (top page 18). We have added a few sentences on this on Page 22.
- The authors have used five treatment groups in the proteomic analysis but the presentation was done for two groups at one time. Also the statistical test was student t test. It will be better if the authors combined all the groups in one figure and use some strong statistical test.
We have now added an additional two-way ANOVA analyses (with post-hoc unpaired t- tests) to compare key targets from the hypoxia and OGD experiments (see new Figure 9). These targets were ACTB, HDAC6 and Uqcrh for hypoxia-treated slices, to further explore the changes in actin, cell stress and mitochondrial signalling. Targets chosen to further explore the identified targets in OGD were CAMK2A, RAC5C and Ctsb, to further examine synaptic plasticity, mitochondrial and apoptosis changes. Two-way ANOVA analyses were conducted separately on hypoxia and OGD comparisons, as these were conducted by separate experimenters rather than all of the groups combined in one ANOVA analysis. The pathway analyses utilised Fishers exact test with the Benjamin-Hochberg correction for multiple comparisons applied to decrease the chance of spurious conclusions. This information has been added to section 2.4.4
Reviewer 2 Report
Numerous minor formatting issues need to be addressed, and the manuscript needs to be reviewed carefully for similar errors for I have not read the entire manuscript at the level of detail to catch all of these. Some examples: For superoxide, the radical symbol and negative charge should be superscripts. "ROS" is not defined at first use. "Glucose" should not be capitalized. A space should be placed between a number and the corresponding unit every time. No space between "manganese" and "(III)". The use of first person voice, e.g. "we", should be limited to the introduction and conclusion. The first sentence on page 19 referring to previous results lacks a reference. In addition, a number of very minor grammatical issues are present.
Some animal details are missing. What was the source of the animals? What was the animals' diet?
The caption to Table 1 should be on the same page as the table.
The figure caption to Figure 2, part F is nonsense. The figure has only grey and green bars while the text describes white and black bars.
Are these z-scores really good to 10 significant figures???
I have no problems with the experimental approach and interpretation of the results. The conclusions are sound. The manuscript is generally well written. My issues are all in careful attention to details of presentation.
Author Response
Referee 2
Numerous minor formatting issues need to be addressed, and the manuscript needs to be reviewed carefully for similar errors for I have not read the entire manuscript at the level of detail to catch all of these. Some examples: For superoxide, the radical symbol and negative charge should be superscripts. "ROS" is not defined at first use. "Glucose" should not be capitalized. A space should be placed between a number and the corresponding unit every time. No space between "manganese" and "(III)". The use of first person voice, e.g. "we", should be limited to the introduction and conclusion. The first sentence on page 19 referring to previous results lacks a reference. In addition, a number of very minor grammatical issues are present.
We have now had the manuscript re read thoroughly and have made all these changes.
Some animal details are missing. What was the source of the animals? What was the animals' diet?
The rats were sourced from Charles River and bred in the Biomedical Facility in UCD. Animals diet was a Global Diet 2918. These details are now in the methods section 2.1.
The caption to Table 1 should be on the same page as the table.
This has now been placed on the same page.
The figure caption to Figure 2, part F is nonsense. The figure has only grey and green bars while the text describes white and black bars.
Apologies, part F) has now been rewritten.
Are these z-scores really good to 10 significant figures???
The referee is absolutely correct and we now give all this data to two decimal places.
I have no problems with the experimental approach and interpretation of the results. The conclusions are sound. The manuscript is generally well written. My issues are all in careful attention to details of presentation.
We have carefully reviewed the Manuscript.
Reviewer 3 Report
Dear Authors,
The manuscript titled "An electrophysiological and proteomic analysis of the effects of the superoxide dismutase mimetic, MnTMPyP, on synaptic signalling post ischemia in isolated rat hippocampal slices" is very interesting, well written and organized. However, in order to improve the quality of the paper I have few minor suggestions.
1- In the "2.4 Drugs" subheading, please indicate the stock solution used in order to know the DMSO vehicle concentration the was applied for the experiments.
2- Did the Authors evaluate also other antioxidant molecules such as Vitamin E, or CBD that have a role both in synaptogenesis and in antioxidant defence?
3- Concerning the MnTMPyP treatment, did the authors perform pre-treatment in order to evaluate the preventive effect of the SOD mimetic?
4- Please, use the same abbreviation for indicating the hours, such as "h" or "hr".
Author Response
Referee 3
The manuscript titled "An electrophysiological and proteomic analysis of the effects of the superoxide dismutase mimetic, MnTMPyP, on synaptic signalling post ischemia in isolated rat hippocampal slices" is very interesting, well written and organized. However, in order to improve the quality of the paper I have few minor suggestions.
1.In the "2.4 Drugs" subheading, please indicate the stock solution used in order to know the DMSO vehicle concentration the was applied for the experiments.
(Now Section 2.5) MnTMPyP was dissolved in dimethyl sulfoxide (DMSO) with a final concentration equal or less than 0.1% when diluted in aCSF. DMSO control experiments were carried out at the same dilution. This has now been added to section 2.5, Page 5.
2.Did the Authors evaluate also other antioxidant molecules such as Vitamin E, or CBD that have a role both in synaptogenesis and in antioxidant defence?
We are currently testing the effect of the free radical scavenger edaravone, in electrophysiological experiments and propidium iodide (PI) cell viability assays in an OGD model. Edaravone is a potent antioxidant compound and has previously been reported to have a neuroprotective effect reducing apoptosis and stimulating the BDNF/TrkB pathway, crucial in neural development, neurogenesis and synaptic plasticity (Yang. Y., 2022, Li. Q., 2022). Our preliminary results do not show significant changes in fEPSP during a 20 min OGD protocol in both CA1 and DG regions of the hippocampus. However prior treatment of organotypic hippocampal slices exposed to a 1 hr OGD insult with edaravone, showed improved cell viability up to 24 hr. We now mention some of these preliminary effects in the Discussion, Page 24.
3- Concerning the MnTMPyP treatment, did the authors perform pre-treatment in order to evaluate the preventive effect of the SOD mimetic?
In section 2.2. we have added “Slices were incubated with MnTMPyP (25 µM and 2.5 µM) for 20 min before switching to ischemic condition and LTP induction.” In section 2.3, we have added “ Slices were incubated with MnTMPyP (25 µM and 2.5 µM) for 20 min before performing hypoxia or OGD.”
4- Please, use the same abbreviation for indicating the hours, such as "h" or "hr".
We have now re read the manuscript carefully and corrected all abbreviation issues.
Round 2
Reviewer 1 Report
The authors have addressed all my concerns in the revised manuscript and also added the limitation of the research. Now the quality of the manuscript has significantly approved. I support the publication of the revised manuscript.